# Endothelial Dysfunction in Neurodegenerative Diseases

**DOI:** 10.3390/ijms24032909

**Published:** 2023-02-02

**Authors:** Yao-Ching Fang, Yi-Chen Hsieh, Chaur-Jong Hu, Yong-Kwang Tu

**Affiliations:** 1Taipei Neuroscience Institute, Taipei Medical University, Taipei 11031, Taiwan; 2Graduate Institute of Neural Regenerative Medicine, College of Medical Science and Technology, Taipei Medical University, Taipei 11031, Taiwan; 3Department of Neurology, Shuang Ho Hospital, Taipei Medical University, New Taipei City 23561, Taiwan

**Keywords:** cerebral blood flow (CBF), blood–brain barrier (BBB), Alzheimer’s disease (AD), vascular dementia (VaD)

## Abstract

The cerebral vascular system stringently regulates cerebral blood flow (CBF). The components of the blood–brain barrier (BBB) protect the brain from pathogenic infections and harmful substances, efflux waste, and exchange substances; however, diseases develop in cases of blood vessel injuries and BBB dysregulation. Vascular pathology is concurrent with the mechanisms underlying aging, Alzheimer’s disease (AD), and vascular dementia (VaD), which suggests its involvement in these mechanisms. Therefore, in the present study, we reviewed the role of vascular dysfunction in aging and neurodegenerative diseases, particularly AD and VaD. During the development of the aforementioned diseases, changes occur in the cerebral blood vessel morphology and local cells, which, in turn, alter CBF, fluid dynamics, and vascular integrity. Chronic vascular inflammation and blood vessel dysregulation further exacerbate vascular dysfunction. Multitudinous pathogenic processes affect the cerebrovascular system, whose dysfunction causes cognitive impairment. Knowledge regarding the pathophysiology of vascular dysfunction in neurodegenerative diseases and the underlying molecular mechanisms may lead to the discovery of clinically relevant vascular biomarkers, which may facilitate vascular imaging for disease prevention and treatment.

## 1. BBB Function

The brain, which lacks a long-term energy storage system, has a high demand for energy and depends on the continuous supply of oxygen and nutrients. The cerebral vascular system is normally responsible for homeostasis and tightly supervises CBF; however, its maladaptation hinders normal tissue and body functions. The unique anatomy and physiology of the brain make it an immune-privileged organ; for example, the cerebral blood vessels have specific anatomical as well as functional characteristics and constitute the BBB, which is a functional barrier that prevents plasma proteins from entering the parenchyma [1]. The BBB serves as a bridge between peripheral circulation and the central nervous system (CNS). BBB microvessels are structurally different from peripheral blood vessels. The microvessels contain only a single layer of endothelial cells (ECs) and a large number of mitochondria; they lack perforated fenestrations and form tight junctions (TJs).

The BBB also protects the brain from pathogenic infections and regulates the transport of oxygen, nutrients, certain substances, and metabolic wastes. It maintains the ionic balance on both sides of the barrier, facilitates the transport of nutrients, and prevents the influx of harmful molecules. In the absence of specific transport systems or lipophilic molecules, macromolecules cannot diffuse through the BBB. Cerebral ECs, astrocytes, and pericytes form a neurovascular unit (NVU), which maintains BBB functions [2]. The membrane molecules of these cells function as immune surveillants and facilitate leukocyte infiltration during neuroinflammation [3]. Therefore, blood vessel occlusion, inflammation, and infection may affect the cerebrovascular system, which, in turn, affects the entire body. Brain diseases are associated with vascular dysfunction.

Pericytes are involved in the attachment of cerebral blood vessels and the regulation of the growth and movement of ECs. A basal membrane comprising collagen laminin, fibronectin, and other matrix proteins wraps the cerebral ECs to provide support to the BBB and maintain it. Although the neurons and microglia are not part of the NVU, they are located in the close vicinity of ECs and regulate barrier functions [4].

Owing to the TJs forming a physical and functional barrier between ECs, the influx of substances from the blood into the brain is restricted. These TJs comprise occludins, claudins, and junctional adhesion proteins. The scaffolding proteins zona occludens (ZO) interact with occludin and actin. Because the ECs of the BBB contain numerous mitochondria, they are believed to ensure high resistance to nonselective molecular permeation. The resistance of the BBB is approximately 100 times higher than that of other peripheral small blood vessels [5]. The physical barrier is created primarily through the formation of TJs, which restrict the passive diffusion of macromolecules and systemic vascular cells as well as molecules. The membrane of ECs contains numerous transporters, which further strengthen the barrier function. The transporters allow the transport of molecules from the lumen to the ECs and vice versa. Upon entering the BBB ECs, toxic substances are subjected to certain enzymes, such as the cytochrome P-450 system, monoamine oxidase, and others; thus, the BBB effectively protects the brain from the effects of harmful drugs, toxins, and peripheral vascular molecules [6].

## 2. BBB Dysfunction

It is well-known that the BBB can be disrupted in aging and neurodegenerative diseases [7,8,9,10]. AD, a key neurodegenerative disease, impairs the language, memory, and cognitive functions of patients, leading to dementia. AD is characterized by the formation of extracellular fibrillar amyloid-β (Aβ) plaques and intracellular neurofibrillary tangles (NFTs; hyperphosphorylated protein-τ (p-τ)), as well as neurodegeneration. The biomarkers of AD neuropathology can be identified by assessing cerebrospinal fluid (CSF) and postmortem tissues, as well as by performing positron emission tomography (PET) [11]; however, patients with dementia may not exhibit the aforementioned pathological features or may have dementia caused by other factors, which indicates a vital role of vascular lesions in this disease. Cerebrovascular disease is a common coexisting comorbidity in AD and may lead to cognitive impairment and dementia (VaD), in addition to presenting with other symptoms; it may lower the threshold value to develop dementia. Studies on structural changes in the microvascular bed have suggested that vascular dysfunction leads to AD-related neuropathological phenomena and cognitive impairment; however, the underlying mechanism remains to be clarified. Currently, researchers believe that neuropathology and blood vessel dysfunction together enhance AD symptoms. Studies have shown that both normal aging and neurodegenerative diseases involve BBB breakdown. Biomarkers associated with BBB breakdown have also been reported [7,12]. Thus, BBB breakdown may help to diagnose aging-related cognitive impairment and dementia.

### 2.1. Aging

Aging leads to BBB breakdown, which increases its permeability; thus, toxic and inflammatory substances enter the brain through blood transmission, leading to neurodegenerative diseases and dementia [9,13]. In older adults, the impairment of the BBB has been demonstrated to be closely associated with AD and cognitive dysfunction [9,14,15,16]. The frequency of neurodegenerative diseases increases with increasing age. AD, Parkinson’s disease, stroke, and various other neurological diseases are prevalent in older adults [17,18]. BBB damage also impairs the influx of nutrients (e.g., glucose) and oxygen, in addition to the efflux of waste products [8,19,20]. The disruption of the BBB can be regarded as a biomarker of the normal aging process [21,22]. Older adults with cognitive impairments are more susceptible to BBB damage than those without them [9].

### 2.2. VaD and AD

The BBB plays a crucial role in maintaining the surrounding microenvironment. Damage to any cellular or molecular component of the BBB may lead to various neurodegenerative diseases, such as AD and VaD [2,13,23]. Dementia, a progressive neurological disorder associated with cognitive impairment and brain function deterioration, affects individuals’ activities of daily living, thereby posing a burden on society [24]. In patients with dementia, hypoperfusion and BBB dysfunction result in brain damage, brain volume reduction, and cognitive impairment [25,26,27]. VaD and AD are two major types of dementia. AD is the most common type of dementia (approximately 70%) among elderly people, whereas VaD is the second most common type (approximately 15%) [25,28,29]. A study published in 2018 reported that approximately 50 million people worldwide had dementia, which was projected to see a 1.6-fold increase by 2030 and a 3-fold increase by 2050 [30].

Cerebral hemorrhage, ischemia, and hypoxia may cause VaD [27]. VaD is a neurological disorder that is characterized primarily by cognitive impairment resulting from reduced CBF due to vascular system damage [27,31]. Aging leads to the thickening of blood vessel walls and the acceleration of vascular turbulence, resulting in BBB damage [13] which, in turn, exacerbates neurological diseases. BBB damage in AD leads to Aβ deposition along the cell wall and the elicitation of inflammatory responses. The accumulation of NFTs in the neuronal cytoplasm is also associated with AD [25,28]. In AD, decreased Aβ clearance has been associated with decreased CBF and cognitive impairment [29]. Aβ causes inflammation, neuronal degeneration, and cognitive impairment, which are associated with BBB damage [15,27,29]. Both vascular and genetic pathways may be responsible for dementia through BBB damage. In an Aβ-independent pathway, a damaged BBB and reduced CBF lead to vascular hypoperfusion and nutrient release, respectively, both of which result in neurodegeneration. In contrast, in an Aβ-dependent pathway, BBB damage and vascular hypoperfusion lead to a decreased clearance of Aβ and amyloid precursor protein (APP), respectively, resulting in their accumulation in the brain. This leads to the formation of NFTs and the elicitation of inflammatory responses, which accelerate neurodegeneration and eventually dementia [13].

Chronic hypoperfusion and thrombosis are key events in VaD; they decrease CBF and increase levels of oxidative stress as well as inflammatory molecules. These chronic events cause damage to the periventricular WM, basal ganglia, and hippocampus. Thus, pathological changes in the cerebrovascular system lead to VaD through brain damage. Vascular damage includes large-vessel atherosclerosis, small-vessel atherosclerosis, and cerebral amyloid angiopathy; these cerebrovascular lesions promote gray matter microinfarction, WM lesion formation, and microbleed formation [32,33,34] throughout the brain, ultimately leading to VaD.

## 3. BBB Breakdown in VaD

Bacterial infection, viral invasion, ischemia, and neurodegenerative diseases can cause BBB damage. Therefore, the BBB may not be a static structure but rather a dynamic structure that is affected by various factors and signals from the brain parenchyma and peripheral blood. In some cases, damage reversal can help achieve therapeutic effects; for instance, the use of the osmotic diuretic mannitol facilitates the paracellular permeation of chemotherapeutic drugs. The deregulation of TJ proteins may also enhance the permeability of the BBB. In earlier studies, mice lacking claudin-5 exhibited increased paracellular permeability [35,36]. Current studies are focusing on the causal relationship between the aforementioned local phenomena and alterations in BBB transporter structures. Given the dynamic and heterogeneous structure of the BBB, BBB damage is likely responsible for the damage to some areas of the brain in patients with degenerative diseases, including VaD. 

With aging, Aβ clearance becomes less effective or Aβ production is increased, which leads to the formation of Aβ deposits in blood vessels. Normally, transporters eliminate excess amounts of Aβ; however, in AD, the dysregulation of the transporter protein leads to the accumulation of Aβ. This accumulation induces microglia to produce reactive oxygen species (ROS), complement proteins, and cytokines for cellular damage; however, the cause of VaD remains unknown. 

## 4. BBB Alterations in AD

The genetic variation across patients with AD may represent a key aspect. The variants of the APP genes presenilin 1 and 2 are associated with familial AD. Mutations in apolipoprotein E (ApoE), a cholesterol transporter, have been associated with late-onset AD [37,38,39]; however, the underlying mechanisms remain obscure. BBB damage is regarded as a hallmark of AD [13,40]. The supposed mechanism underlying BBB damage involves Aβ deposition in the vascular bed, leading to inflammation and neurotoxicity. In animal models of AD, in addition to the changing functions of Aβ transporters, a reduction in the levels of low-density-lipoprotein-receptor-related proteins (LRPs; possible efflux transporters of Aβ) has been noted. Aβ may decrease claudin-5. These findings reveal that Aβ reduces the levels of TJ proteins, thus damaging the BBB. Despite the operation of the widely recognized neurovascular pathology in AD, the BBB sometimes remains intact.

Neuroinflammation plays a key role in the progression of AD, possibly by affecting the BBB. Aβ induces the release of cytokines from microglia, leading to damage to the brain. These cytokines reportedly cause BBB damage, resulting in the further accumulation of amyloids; increased amyloid accumulation leads to the activation of microglia, which, in turn, results in the increased production of cytokines, some of which attract immune cells to the brain. Neuroinflammation-induced barrier disruption may disrupt the immune-privileged state of the brain, inducing peripheral immune responses responsible for disease progression. Although Aβ-induced inflammatory responses and neuronal damage have not been thoroughly studied, the involvement of the peripheral immune system has been demonstrated. In AD, amyloid plays a role in complement activation. The complements have been detected in the cell membranes of damaged neurons [41,42]. These findings highlight the occurrence of peripheral immune-to-inflammatory responses in the brain, which indicate BBB damage. Cytokines regulate EC membrane proteins, which enable immune cells to enter the brain. Tumor necrosis factor (TNF)-α, interleukin (IL)-1β, and interferon (INF)-γ increase the expression levels of intercellular adhesion molecule (ICAM)-1 by activating nuclear factor (NF)-κB, thus allowing leukocytes to migrate and cross the BBB to enter the brain. In ECs, TNF-α increases the expression levels of ICAM-1, vascular adhesion molecule (VCAM)-1, and E-selectin mRNAs. ICAM-1 is also expressed in amyloid plaques, where it attracts immune cells and induces inflammatory responses.

Cytokines and certain growth factors can alter the BBB transporters in ECs. In patients with AD, the levels of glucose transporters are decreased in the cortex and hippocampus of the brain. P-glycoprotein variation has been demonstrated in patients with AD. Because the normal function of P-glycoprotein is the clearance of amyloids from ECs, functional alteration or disruption leads to the excessive accumulation of amyloid plaques, which favors AD pathogenesis.

Many molecules with angiogenetic functional properties may be involved in the pathophysiology of AD. Among them, vascular endothelial growth factor (VEGF) is crucial for maintaining the function of blood vessels, implying the involvement of the pathogenic angiogenesis of cerebral blood vessels. Furthermore, increased numbers of new and total blood vessels have been noted in the postmortem brain tissues of patients with AD. These new, immature blood vessels result from EC growth and movement; however, the vessels have many fenestrations, which compromise BBB integrity by creating an avenue for toxic substances to enter the brain. Notably, AD symptoms can be improved through innovative treatments. Drugs may enter the brain via the BBB through immature blood vessels, abnormal TJs, and altered transporters.

## 5. Cells Maintaining BBB Integrity

The BBB is formed by ECs surrounded by astrocytes, pericytes, microglia, oligodendrocytes, and neurons [15,25,29]. Among them, pericytes, astrocytes, and ECs constitute the NVU [25,29] (Table 1).

### 5.1. Pericytes

ECs grow and maintain BBB integrity together with pericytes, astrocytes, neurons, and glial cells. The degradation of pericytes [27] and a reduction in their number [43] have been observed in aged mice. In aged brain tissues, the increased activity of caspase 3/7 promotes apoptosis and reduces cell viability, thus decreasing the number of BBB pericytes [43]. By contrast, an increase in the number of BBB pericytes was detected in aged rats [44], whereas the number remained unchanged in the brain tissues of aged monkeys [45]. Because pericytes maintain the BBB, the decrease in their number and density in the cortex and hippocampus of patients with AD [46] may subsequently increase the expression levels of Aβ and p-τ, as well as BBB breakdown [47]. The loss of pericytes results in the degradation and increased permeability of the BBB [48]. In VaD, studies have also shown that BBB disruption due to pericyte degeneration leads to the loss of further CBF myelin damage, axonal degeneration, and oligodendrocyte loss [49]. Pericyte deficiency may also upregulate VEGF expression, leading to leaky and hemorrhagic vessels [50]. The dysfunction of pericytes is involved in aggravating dementing diseases, such as AD and vascular dementia [47,49,51]. 

### 5.2. Astrocytes

Astrocytes are essential for maintaining BBB functions. The endfeet of astrocytes are attached to the vessel wall, and nutrients can be exchanged between ECs and the brain parenchyma. The substances released by these cells exert varying effects on the BBB. Astrocytes are key components of the vascular bed and crucial for the integrity of the BBB. The astrocytes release TGF-β, glial cell line-derived neurotrophic factor, fibroblast growth factor -β, and IL-6 to increase the expression levels of endothelial tyrosine kinase receptors and transport proteins (e.g., P-glycoprotein and GLUT-1). These factors facilitate the formation of TJs of ECs. With increasing age, the coverage of blood vessels by astrocytic endfeet and the expression of aquaporin-4 (AQP4) decrease, whereas the expression of glial fibrillary acidic protein increases, thus causing astrogliosis. A study conducted using conditional knockout mice revealed that the deletion of astrocytic laminin led to a decrease in the level of AQP4, ultimately resulting in the loss of TJs in ECs [52]. A study found that the endfoot disruption of astrocytes with a concomitant reduction in AQP4 reduced protein levels of Kir4.1 and potassium channels in a model of hyperhomocysteinemia (HHcy) with vascular cognitive impairment [53]. During aging, oxidative stress promotes the release of cytokines and chemokines from astrocytes. Tissue-type plasminogen activator (tPA) binds to LRP-1 on astrocytes, thereby activating plasmin-mediated Rho kinase and the retraction of astrocyte endfeet from the vessel wall, leading to BBB dysfunction [54]; however, studies have shown that hypoxia leads to the subcellular relocalization of AQP4 to the surface of the BBB and blood–spinal cord barrier (BSCB), accompanied by increasing water permeability via a signaling pathway associated with calmodulin (CaM). In vivo evidence has shown that the inhibition of AQP4 subcellular localization to the BSCB reduces spinal cord water content, the extent of subsequent injury, and enhances recovery in a rat spinal cord injury model of CNS edema [55]. Moreover, targeting CaM to inhibit the cerebral edema mediated by AQP4 may provide a therapeutic target during the acute phase in a photothrombotic stroke model [56].

In the brain tissues of patients with AD, an altered morphology of astrocytes accelerates BBB breakdown [57]. Astrocytes help maintain the BBB by increasing the expression levels of cladulin-5 and occludin [58]; however, in a mouse model of stroke, ischemia-induced astrogliosis reduced the expression levels of the TJ proteins claudin and occludin [58], suggesting the involvement of astrocytes in the regulation of TJ proteins. A loss in astrocytes via tamoxifen causes a decrease in the expression of the TJ protein ZO-1, suggesting that astrocytes are crucial for BBB integrity. 

Furthermore, the depolarization of astrocyte terminals may reduce BBB integrity [59]. In AD models, morphological changes were observed in astrocyte ends located in the close vicinity of Aβ deposits in blood vessels [60]. VaD induced astrocyte and microglial activation, brain injury, and neuronal death in the hippocampus via reducing BDNF neurotrophic factor expression [61].

### 5.3. Microglia

Microglia are widely distributed in the CNS and are activated during the processes of aging and neurodegenerative disease pathogenesis [62,63]. Quiescent microglia maintain a ramified structure; however, during aging and neurodegenerative disease pathogenesis these cells are activated and assume an ameboid structure exhibiting a phagocytic morphology [64]. The inactivation of astrocytes leads to the activation of microglia [65]. The production of ROS accelerates BBB dysfunction and neurodegeneration [66,67]. During aging, oxidative stress activates microglia to release various cytokines (e.g., IL-6, IL-1β, and TNF-α), thus increasing the levels of ROS and reactive nitrogen species, which leads to BBB collapse [68,69,70] and subsequent cellular damage as well as neurodegeneration [71]. In regard to physiological conditions, a mouse study suggested that microglia stimulate the expression of the TJ protein claudin-5 and maintain the integrity of the BBB [72]. In the brain tissues of patients with AD, activated microglia secrete these inflammatory cytokines because of the accumulation of Aβ [73], thus causing BBB damage [74]. Moreover, Abeta induced lymphocyte function-associated antigen 1 (LFA-1), regulating neutrophil extravasation into the CNS, and induced neutrophils passing through the BBB [75]. A loss in microglia via CSF1R inhibition prevents memory impairment in Ang II-induced hypertensive mice, implying that this detrimental effect is mediated by the activated microglia within the BBB [76]. 

### 5.4. Neurons

Neurons are directly connected to astrocytes. The end feet of astrocytes cover blood vessels, where ECs are located; the distribution of the three types of cells indicates a balanced association [63,77]. When this association is dysregulated, the BBB breaks down, resulting in increased permeability [78]. Once BBB integrity is compromised, proteins such as fibrinogen and tPA in the blood cross the BBB and form fibrin aggregates in the brain, resulting in neuronal loss, axonal retraction, and neurodegeneration [79]. A mouse study demonstrated that a fibrinogen-induced loss in spine and cognitive impairment via microglial CD11b, associated with neural damage, are involved in AD [80]. Moreover, moxibustion could improve the disability of learning memory in VD rats via upregulating the numbers of DCX-positive cells for neuronal repair as well as regeneration and downregulating the expression of inflammatory factors [81].

### 5.5. ECs

Under normal conditions, most complement proteins produced by ECs do not cross the BBB [82,83]; however, in the case of CNS damage or BBB dysfunction, ECs release complement regulatory proteins [84] that infiltrate the brain and then activate microglia, oligodendrocytes, and neurons [85]. Endothelial cells also produce C3a and C5a, thus substantially affecting the infiltration of inflammatory cells into the brain and the cascade of cytokines, which ultimately lead to neurodegeneration [86,87]. In neurodegenerative diseases, a complement, tPA, and fibrinogen pass through ECs and lead to neural degeneration as well as impairment. Firstly, Aβ binds to C1q, thus activating complement signaling. The inhibition of the C5/C5aR1 pathway protects against neural damage in AD [88]. Secondly, tPA binds and activates low-density LRP-1 in ECs, followed by the production of substances such as metalloproteinases [89,90,91]. Furthermore, tPA converts inactive plasminogen into active plasmin [92,93], which, in turn, also activates MMPs, leading to the degradation of TJs [94,95]. Thirdly, fibrinogen activates the microglia, induces ROS production, and leads to a loss in oligodendrocytes [80]. In patients with chronic cerebral hypoperfusion, considerable increases in the levels of ICAM-1 as well as VCAM-1 in vascular ECs have been associated with cognitive impairment [96,97]. A single-nucleus transcriptome analysis has shown that VaD contains an EC subcluster that expresses genes associated with programmed cell death [98].

### 5.6. Oligodendrocytes

Damage to oligodendrocytes results in the inhibition of remyelination [99]. Demyelination hinders the transmission of neural signals, leading to cognitive impairment. Biological stresses, such as hypoxia, oxidative stress, and inflammation, lead to impaired neurogenesis, impaired neuronal precursor cell proliferation, a decrease in synaptic plasticity, and a decrease in the density of neural spines, resulting in cognitive impairment [100,101]. Fibrinogen phosphorylates SMAD and causes oligodendrocyte progenitor cell (OPC) repression, leading to oligodendrocyte loss, followed by neurodegeneration [102]. In a rat model of small-vessel disease, dysfunctional ECs block oligodendrocyte differentiation via heat shock protein 90α, leading to impaired myelination and dementia [103].

### 5.7. Macrophages

Perivascular cells do express CD163, but microglia do not [104]. They were different from microglia. PVMs are macrophages located in the perivascular Virchow–Robin space (VRS) of the CNS. Under physiological conditions, perivascular macrophages act as the first “firewall” in the CNS to help maintain TJs and reduce vascular leakage, pathogen load, and inflammation [105] because of their innate immune functions. Furthermore, these macrophages contain scavenger receptors, which may help remove toxin products from the brain parenchyma [106] and play various roles in the pathophysiology of diseases such as AD [105]. A growing number of studies has suggested that, besides microglia, the phagocytosis of PVMs is involved in Aβ clearance. A loss in the function of PVMs is related to the vascular accumulation of Aβ 42 and the severity of cerebral amyloid angiopathy [104]. M1 macrophage subset increment and M2b macrophage subset decrement were observed in AD patients with cognitive impairment compared with controls [107]. Microglia/macrophages with scavenger receptors (SRs) could bind endogenous and foreign molecules, resulting in the phagocytosis of these cells. 

### 5.8. Fibroblasts (FBs)

Changes in the functions of perivascular FBs have been associated with other neurological disorders [108]. These cells express *LAMA2*, *LAMB1*, and *LAMC1*. These LAMAs encode laminin, which interacts with dystrophin and regulates AQP4 in astrocytes; any damage to FBs surrounding blood vessels results in the dysregulation of AQP4 associated with the glymphatic system, which may facilitate Aβ aggregation and AD [109]. Time-lapse imaging data showed that perivascular fibroblasts could be pericyte precursors. Perivascular fibroblasts with genetic variation markedly reduce collagen deposition around endothelial cells, resulting in grotesque blood vessels. Perivascular FBs express the extracellular matrix genes *col1a2* and *col5a1*, which are believed to regulate vascular integrity. In an earlier study, zebrafish lacking *col5a1* exhibited additional spontaneous bleeding, which suggests that perivascular FBs stabilize vascular integrity [110]. They construct the ECM surrounding blood vessels for stabilization and then play roles as pericyte progenitors to differentiate pericytes that contribute to vessel integrity.

**Table 1 ijms-24-02909-t001:** Cells of the BBB in aging, Alzheimer’s disease (AD), and vascular dementia (VaD). (BBB: blood–brain barrier, AD: Alzheimer’s disease, VaD: vascular dementia, and N/A: not applicable).

Cells	Disease	Cell Number	Phenotype	Reference
Pericyte	Aged mice	Decrease	BBB degradation	[27]
	Aged mice	Decrease	BBB reduction	[43]
	Aged brain	Decrease	Cell numbers decreased in the BBB	[44]
	Aged brain	Decrease	Cell numbers decreased in the BBB	[111]
	Aged monkeys	Unchanged	Cell numbers unchanged	[45]
	AD	Decrease	Amyloid beta and p-tau proteins increased	[47]
	AD	Decrease	BBB degradation	[111]
	Aged rats	Increase	Cell numbers increased	[44]
	VaD	Decrease	Pericyte dysfunction	[49]
Astrocyte	Aging	N/A	BBB dysfunction	[54]
	Aging	N/A	CLDN5 and OCLN increased	[58]
	Aging	N/A	CLDN5 and OCLN increased	[57]
	Aging	N/A	TJ proteins, claudin-5, and occludin decreased	[62]
	KO mice (deletion of astrocytic laminin)	N/A	Loss in TJs in ECs	[52]
	AD brain	N/A	BBB breakdown	[47]
	AD brain	N/A	Depolarization of astrocyte terminals	[59]
	AD	N/A	Morphological changes in astrocyte ends	[60]
	VaD	Cell activation	Brain injury, lipid peroxidation, and neuronal death	[61]
Microglia	Aging /neurodegenerative diseases	N/A	Microglia activated	[62]
	Aging /neurodegenerative disease	N/A	Became an amoeba or phagocytic morphology	[64]
	Aging /neurodegenerative disease	N/A	Leakage of the BBB	[10]
	Aging	N/A	BBB collapsed	[69]
	Aging	N/A	BBB collapsed	[70]
	Altered microglia morphology	N/A	BBB integrity compromised	[72]
	AD brain	N/A	Secrete inflammatory cytokines by microglia	[73]
	AD brain	N/A	BBB damage	[74]
	Hypertension	Cell activation	Increased permeability of the blood–brain barrier (BBB)	[76]
Neuron	Production of reactive oxygen species	N/A	BBB dysfunction	[74]
	Production of reactive oxygen species	N/A	Neurodegeneration	[67]
	BBB integrity compromised	N/A	Fibrin aggregates in the brain	[79]
	Accumulation of fibrin in the brain	Decrease	Cause damage to neuronal axons	[80]
	VaD	Increase	The number of DCX-positive neurons increases	[81]
Endothelial cells	Plasmin activate MMPs	Decrease	Degradation of TJs and basal lamina	[94]
	CNS damage/BBB dysfunction	Decrease	ECs release complement regulatory proteins, which infiltrate the brain	[85]
	Produced C3a and C5a binding to C3aR and C5aR1	N/A	Infiltration of inflammatory cells into the brain	[86]
	Produced C3a and C5a binding to C3aR and C5aR1	N/A	Cytokine cascade	[87]
	AD	N/A	Beta-amyloid (Aβ) activates complement signaling by binding to C1q	[88]
	AD	N/A	Inhibition of the C5/C5aR1 pathway protects against damage	[88]
	Cognitive impairment	N/A	Increased ICAM-1 and VCAM-1 in vascular ECs in CCH	[97]
	VaD	Decrease	Express genes associated with programmed cell death	[98]
Oligodendrocyte	Damage to oligodendrocytes	N/A	Inhibition of remyelination	[99]
	Hypoxia, oxidative stress, and inflammation	N/A	Cognitive impairment	[101]
	SVD	N/A	Block oligodendroglial differentiation	[103]
Macrophage	Perivascular macrophages (PVMs)	N/A	Reduce vascular leakage	[105]
	PVMs	N/A	Lower pathogens	[105]
	PVMs	N/A	Limit inflammation	[105]
	PVMs	N/A	Remove toxin products from the brain parenchyma	[106]
	Deficiency of CD36 and Nox2 in macrophage	N/A	Inhibited ROS production	[106]
	AD	N/A	M2b macrophage subset decrement and M1 macrophage subset increment	[107]
Fibroblast (FB)	AD	N/A	Phagocytose and alleviate Aβ plaques	[105]
	Zebrafish lacking col5a1,	N/A	Under genetic ablation of the col1a2 gene, additional spontaneous bleeding	[110]
	Aβ aggregation and AD	N/A	Damage to FBs around blood vessels, leads to the dysregulation of AQP4	[109]
	Neurological disorders	N/A	Altered activity of perivascular FBs	[108]

## 6. Mechanisms Underlying BBB Breakdown in AD and VaD 

BBB ECs are consecutively and tightly joined together through TJ proteins, such as claudins, occludins, and ZO-1, which conjugate with actin to limit intercellular macromolecular diffusion in the CNS. Toxins and wastes in the brain parenchyma are excreted through transporters (Figure 1). 

Hypoxia, oxidative stress, and inflammation cause BBB breakdown in AD and VaD. Hypoxia results in increased levels of oxidative stress, which lead to the production of peroxynitrite ions, ROS, and free radicals [112,113,114]. Moreover, through the use of gene expression microarrays and a proteome profiling array, hypoxia was found to induce the Wnt- and p53-related apoptotic signaling pathways in primary human astrocytes, implying that BBB breakdown could be via apoptosis. [115]. Oxidative stress further disrupts the antioxidant–ROS ratio and damages ECs, glial cells, and neurons, thus causing BBB damage [116]. Cerebrovascular hypoxia also produces inflammatory molecules, which are involved in apoptosis and impaired microvascular function. These inflammatory molecules damage ECs, glial, and neuronal cells, increasing BBB permeability [117]. The inflammatory cytokines, such as TNF-α, IL-1, IL-6, or molecules, such as MMPs (MMP-2 and MMP-9), can infiltrate the brain, cause demyelination, and damage axons as well as oligodendrocytes, facilitating the development of hippocampus and WM lesions [117,118,119,120,121].

## 7. Similarities and Differences between AD and VaD

Studies have revealed differences between AD and VaD in terms of clinical symptoms [122,123]; however, the two disorders sometimes co-occur, a condition known as mixed disorder [124]. In mixed dementia, vascular lesions modulate AD progression and AD-related lesions enhance vascular damage [125]. Genetically, *ApoE* is associated with both vascular dysfunction and AD in older individuals, which suggests that both disorders share a similar pathway [126].

AD and VaD have some similarities; for example, both disorders share the clinicopathology of cognitive impairment and behavior-related changes in psychiatric symptoms [127]. Neuropsychiatric disorders with frontal lobe and white matter hyperintensity have been associated with AD, VaD, and mixed AD/VaD [128]. Epidemiological evidence has revealed mixed neuropathological lesions in AD and VaD [129]. Thus, considerable overlaps are apparent between AD and VaD’s pathophysiology. Therefore, the pathophysiology of all types of dementia can be explored concurrently, rather than having separate investigations for different dementia types, such as AD and VaD.

## 8. Factors Affecting the BBB

Transporters, receptors, and other proteins produced by ECs and AQP4, produced by astrocytes, regulate BBB homeostasis. The deregulation of these proteins may lead to BBB dysregulation (Table 2).

### 8.1. Transporters and Receptors

LRP1 is a major receptor that helps remove Aβ fibrils from the parenchyma. The level of LRP1 decreases with increasing oxidative stress [130,131], which hinders Aβ removal from the brain, increasing Aβ accumulation [132,133]. In mice, lipopolysaccharide-induced systemic inflammation and decreased LRP-1 as well as Pgp levels have been observed; a reduction in the levels of both transporters hinders amyloid clearance from the brain [134,135]. Receptor for advanced glycation end products (RAGE), an immunoglobulin superfamily of cell surface receptors, serves as a receptor for Abeta on endothelial cells, astrocytes, neurons, and microglia [136,137,138]. RAGE, as opposed to LRP1, helps transport Aβ from circulation to the brain, promoting neuronal inflammation, which shows RAGE as a potential therapeutic target. 

GLUT1 deficiency in AD mice leads to behavioral deficits, cerebral microvascular degeneration, blood flow reductions, BBB breakdown, and a reduction in glucose levels [139].

### 8.2. Channel Proteins

AQP4 constitutes the major water channel. It is mainly expressed in CNS astrocytes and is involved in the maintenance of BBB stability and nervous system physiology as well as diseases [140]. Specifically, AQP4 promotes the clearance of Aβ, and the dysregulation of AQP4 leads to the accumulation of Aβ in the brain [141,142]. Astrocytes of the NVU also maintain extracellular potassium concentrations in the brain via a process called “potassium siphoning” [143,144]. It is reasonable to predict that changes in potassium channel function may regulate brain function. Water movement through AQP4 water channels located in the terminal feet of astrocytes maintains osmotic balance and promotes efficient potassium siphoning. In animals with AQP4-deficient astrocytes, the efficient removal of amyloid from the brain is impeded [145]. Furthermore, the disruption of AQP4 expression results in Aβ deposition and inflammation in the human brain, leading to AD [146]. In patients with AD, AQP4 levels are decreased in the perivascular space, which is associated with increased NFT levels and enhanced Aβ pathology [147]; however, the protein levels of AQP4, instead of subcellular translocation, were not changed in many scenarios, including hypothermia, via transient receptor potential vanilloid 4 (TRPV4) activation [148], and osmolality changes via PKA phosphorylation in primary human cortical astrocyte [149]. Moreover, the disturbance of dynamic redistribution in AQP4 could affect the process and synaptic activity in astroctyes [150]. Therefore, AQP4 may be a promising therapeutic target in AD and other diseases of the CNS.

There is no effective drug targeting AQP4 that has been directly approved for clinical use, although AQP4 has been proven to have therapeutic potential in stroke, traumatic brain injury, and other neurodegenerative diseases [151,152,153]. Targeting the calmodulin-mediated AQP4 phosphorylation and translocation in astrocytes with an antipsychotic drug, trifluoperazine, ablated CNS edema in a spinal cord injury model, and this provides the potential for specificity to reduce the side effect [55]. There are also differential kinases, such as ERK, p38, and calmodulin-dependent protein kinase, implicated in AQP4 relocalization, but whether AQP4 is phosphorylated by these kinases directly remains to be determined [115,148,154]. Whether their pores are intrinsically undruggable or could be blocked by other small molecules remains to be studied; however, targeting the molecular mechanisms of them provides new strategies for drug discovery. The selective antagonists or blockers of AQP4 water pores has not been proven; however, the AQP4 inhibitor TGN-020 or AER-270 has beneficial effects on ischemia-induced brain edema [155,156]. The protective effect of these drugs or molecules may have other AQP4 functions independently.

AQP4 is found in endfeet of astrocytes, ventricle lining, tripartite synapses, and the surrounding meninges. A tight correlation between AQP4 and glymphatic function has been obtained from studies on AQP4 or alpha-syntrophin (Snta1) knockout (KO) mice. Previous studies found that the dynamic translocation of AQP4, from intracellular vesicle pools to the plasma membrane, is responsible for water homeostasis in health and disease [150,152,153,157,158]. In pathological conditions, the pharmacological inhibition of AQP4 relocates to astrocyte endfeet to promote edema formation, and Salman et al. proposed that AQP4 is directly involved in this process [158]; however, there is still no conclusion on whether brain water homeostasis is achieved by changing AQP4 itself directly or whether it is an indirect effect caused by the genetic deletion of AQP4 [159]. AQP4 KO mice exhibit a series of physiological and cellular changes, including an increase in brain water content [160], extracellular space in brain tissue [161], the expression of excitatory amino acid transport protein 2 (EAAT2) [162], and a decrease in the expression of anchor proteins, α-syntrophin and dystrophin, in astrocyte terminals [163], which all may affect the expression of other membrane proteins.

### 8.3. ApoE

ApoE is associated with late-onset AD. The function of ApoE is cholesterol transport, and a variant form of ApoE increases beta-amyloid deposition as well as plaque formation in blood vessel walls. The ApoE protein is encoded by *APOE*, which is located on chromosome 9. Three alleles of *APOE* are translated into ApoE2, ApoE3, and ApoE4 isoforms. The ApoE detected is produced by astrocytes in the CNS [164].

ApoE isoforms play differential roles in maintaining the integrity of the BBB [165]. An APOE4 isomer is a major risk factor for AD, and the binding of Aβ to ApoE4 changes the rapid clearance of soluble Ab by LRP1 to its slower clearance by very-low-density lipoprotein receptor [166,167]. APOE4 overexpression reduces BBB integrity by promoting pericyte degeneration in patients with AD [111]; compromised BBB integrity enhances its permeability [130]. The presence of APOE4 increases an individual’s susceptibility to BBB damage; furthermore, the number of pericytes is lower in individuals carrying *APOE4* than in noncarriers [130,168,169]. However, BBB damage was also observed in ApoE^−/−^ mice [170,171,172], suggesting that the function of ApoE is crucial for maintaining BBB integrity. ApoE2 and ApoE3 interact with pericyte LRP-1 to block the cyclophilin-A-NF-κb-MMP-9 (CypA-NF-κb-MMP-9) pathway, thus inhibiting MMPs’ ability to maintain BBB integrity [111]; EC LRP1 knockout results in the activation of the CypA-NF-κB-MMP-9 pathway, leading to TJ and BBB damage [173]. Compared with mice transfected with APOE3 or APOE2, those transfected with APOE4 exhibited increased Glut1 inhibition and RAGE expression [37]. Taken together, the aforementioned findings suggest that ApoE2 and ApoE3 inhibit inflammation by interacting with pericyte LRP-1 and maintain BBB integrity. In contrast, ApoE4 is associated with increased BBB damage or breakdown. The inhibition of ApoE4 or the CypA-NF-κB-MMP9 pathway in patients with AD may be targeted in the future to deaccelerate neurodegeneration. In patients with AD, BBB damage leads to the accumulation of fibrinogen, albumin, thrombin, and IgG around blood vessels; concurrently, TJs are disrupted and ECs as well as pericytes are reduced in number [174]. 

### 8.4. TGF-β

In addition to the formation of extracellular senile plaques and intracellular NFTs, the pathological features of AD include neurodegeneration, cerebral amyloid angiopathy, and cortical blood vessel degeneration. The first two are found in the hippocampal and cortical regions. According to the neuroinflammation hypothesis, the levels of cytokines (TNF-α and IL-1β) and ApoE are elevated in patients with AD, which is associated with AD pathology and disease progression. With inflammation, anti-inflammatory substances are also produced. TGF-β is an immunosuppressive cytokine found in the plaques and NFTs of postmortem tissues; however, in AD transgenic mice, TGF-β, which is abundantly expressed in astrocytes, promotes the amyloidosis of cerebral blood vessels [175]. TGF-β was also found to promote amyloid clearance via microglia [176]. In patients with AD, the level of TGF-1β in CSF is negatively correlated with the level of Aβ [177]; hence, it may be involved in the metabolism of amyloids and the promotion of amyloid deposition as well as clearance.

### 8.5. PDGF-β

Pericytes express PDGF receptor (PDGFR)-β in aged mouse brain tissues. In aged human brain tissues, the levels of PDGFR-β are elevated in CSF, implicating the impairment of pericytes, which is associated with BBB damage [178]. Individuals carrying *APOE4* exhibit more severe cognitive impairment with elevated levels of PDGFR-β in CSF than noncarriers do [179]. Because pericytes are crucial for maintaining the BBB, the dysfunction of pericytic signaling leads to the breakdown of the BBB, thus causing dementia and other neurodegenerative diseases [180]. An increase in the level of soluble PDGFR (sPDGFR) due to the disruption of pericytes and the BBB has been associated with cognitive decline independently of CSF Aβ or tau levels [181]. In Pdgfr +/− pericyte-deficient mice, BBB impairment was demonstrated to result in neuronal degeneration [111]. In a mouse model of AD, pericyte degeneration led to BBB dysfunction, resulting in Aβ accumulation and p-τ phosphorylation [47]. BBB damage associated with pericyte reduction has also been reported in patients with AD [46]. In these patients, BBB leakage begins in the hippocampus and subsequently leads to an increase in sPDGFR-β level in CSF [15,182]. Thus, the level of sPDGFR-β in the CSF can be used as a biomarker of neurodegenerative diseases, such as dementia [178]; however, the role of this molecule is controversial. Cerebral ECs secrete PDGF and activate PDGFR-β signaling, thus contributing to the survival of pericytes [183]. The inhibition of PDGF-BB-induced signaling leads to reduced pericyte recruitment, decreased matrix deposition, and increased vascular hemorrhage [178].

## 9. Radiology

Various imaging techniques and other methods are used to identify BBB-related biomarkers for effective decision making in the treatment of neurodegenerative diseases. Acute BBB injury requires an urgent intervention; however, neuroimaging equipment is not always readily available, and the diagnosis process is relatively long (examination to interpretation). Therefore, blood biomarkers may help resolve the aforementioned problems. 

Dynamic contrast-enhanced (DCE) magnetic resonance imaging (MRI) techniques are available for the direct identification and localization of model-free parameters as well as quantitative parameters with pharmacokinetic modelling [184]. A study with DCE-MRI indicated that aged people with cognitive impairment had higher BBB permeability than healthy individuals [178]. Furthermore, in a study conducted in healthy older adults, the DCE-MRI technique was used with intravenous gadolinium contrast to investigate localized BBB leakage in the brain region, which is the most susceptible to aging-related damage [185]. It could be used in AD patients; the BBB permeability index Ktrans revealed leakage in the hippocampus CA1 and dentate gyrus, implying that Abeta-induced breakdown of the BBB is initiated in the hippocampus. The aforementioned studies confirm that BBB leakage can be detected using DCE-MRI. Using different neuroimaging approaches may help to explore the biomarkers associated with BBB damage in various diseases of the CNS [186], such as increased gadolinium leakage (DCE-MRI), reduced glucose transport (fluorodeoxyglucose-PET), microbleeds (T2-weighted MRI and susceptibility-weighted imaging), or reduced P-glycoprotein-1 function (verapamil-PET).

## 10. Serum Molecules

Secreted protein acidic and rich in cysteine (SPARC), a nonstructural protein of the extracellular matrix (ECM), can reduce transendothelial electrical resistance and TJ proteins, increase paracellular permeability by regulating the tyrosine kinase pathways [187], induce neuroinflammation, and activate the M1 phase of microglia [188]. Therefore, SPARC may be a potential therapeutic target in AD [188]. In individuals with BBB impairment, the levels of P-selectin, E-selectin, and platelet endothelial cell adhesion molecule-1 (PECAM-1) are increased; these can serve as biomarkers of BBB breakdown [189]. Another protein biomarker relevant in this context is s100b, which is produced by astrocyte endfeet; in progressive dementia, the protein levels of a 14:3:3 protein:s100b:phospho-tau ratio was elevated [190]. Recently, a selective TGF-β pathway inhibitor, RepSox, was demonstrated to inhibit VEGF-α expression, and inflammation-related pathways in a CLDN5-P2A-GFP stable cell line [191]. RepSox markedly increases BBB resistance, induces TJs and transporters, reduces paracellular permeability, and prevents VEGF-induced BBB breakdown; hence, RepSox may be used in BBB-targeted therapy for neurological diseases such as AD [191]. 

### 10.1. Acetylcholine Esterase

Current clinically proven drugs that can alleviate AD symptoms are acetylcholine esterase inhibitors. These drugs are also effective for mild dementia in patients with AD; however, they lose their effectiveness with disease progression. Although nonsteroidal anti-inflammatory drugs have not been approved for AD by the Food and Drug Administration, they have been demonstrated to reduce the level of Aβ accumulation in mice.

### 10.2. Antibodies and Chelators

Antibodies for Aβ or targeted molecules with nanoparticles can cross the BBB through their receptors and enter the brain through damaged TJs. Active and passive immunizations with Aβ have been shown to reduce learning and memory impairments in AD mice, microgliosis, neurotic dystrophy, and τ pathology. The accumulation of considerable levels of metals in the brain parenchyma of patients with AD leads to the formation of free radicals and the exacerbation of neurodegeneration; thus, metal chelators can enter the brain through TJs and effectively decelerate AD symptoms. The high lipophilicity and lack of suitable transporters can be overcome by using nanoparticles. Nanoparticles and neuropeptides are currently used for drug delivery. Nanoparticles can target specific proteins involved in neurodegeneration and inflammation, such as ICAM-1, which is expressed in high levels during inflammation and regulates leukocyte attachment as well as extravasation. Nanoparticles can be transported to the inflammation area where they compete with leukocytes to bind to ICAM-1, thereby reducing the extravasation of leukocytes into the parenchyma. Furthermore, nanoparticles bind to integrin αvβ3 or integrin and disrupt angiogenesis. Neuropeptides are small molecules that can enter the brain and react with Aβ when the BBB is compromised; however, because of the associated side effects (mostly severe), the use of neuropeptides is limited. Neuropeptide molecules can enter the brain through BBB cellular junctions, transporters, and cellular uptake.

### 10.3. Malaria

More recently, it has been shown that vascular endothelial cell function is also disrupted by cerebral malaria (CM) infection [192]. A hallmark of endothelial dysfunction is the adhesion of infected RBCs, along with platelets and immune cells, to the vascular endothelium to sequester in the patient’s brain upon infection with the plasmodium parasite. The disruption of BBB integrity and the recruitment of peripheral immune cells are due to iRBC cell adhesion to the brain microvasculature, especially endothelial cells, during the trophoblast and schizont stages, which induce vasospasm. Additionally, changes in the levels of vascular regulatory molecules further promote proinflammatory and prothrombotic states. The sequestration of iRBCs in the brain occurs via cell adhesion molecules, ICAM-1, and endothelial protein C receptor (EPCR), expressed on the surfaces of brain ECs; it induces monocyte to phagocyte iRBCs and then excretes the parasite as well as plasmodium toxins. All of these consequences cause endothelial cell dysfunction, which contributes to the activation of immune cells, axonal damage, neurodegeneration, and cognitive impairment.

**Table 2 ijms-24-02909-t002:** Molecule factors of the BBB in aging, Alzheimer’s disease (AD), and vascular dementia (VaD). (BBB: blood–brain barrier, AD: Alzheimer’s disease, VaD: vascular dementia, CLDN5: claudin-5, OCLN: occludin, TJ protein: tight junction protein, EC: entorhinal cortex, Aβ: amyloid beta, ICAM-1: intercellular adhesion molecule 1, VCAM-1: vascular adhesion molecule 1, CCH: chronic cerebral hypoperfusion, SVD: small-vessel disease, PVMs: perivascular macrophages, ROS: reactive oxygen species, FB: fibroblast, GLUT1: glucose transporter, LRP1: low-density-lipoprotein (LDL)-receptor-related protein 1, Pgp: phosphoglycolate phosphatase, RAGE: advanced glycosylation end product (AGE) receptor, AQP4: aquaporin-4, APOE: apolipoprotein E, VLDLR: very-low-density lipoprotein receptor, CypA-NF-kb-MMP-9: cyclophilin-A nuclear factor κb-matrix metalloproteinase 9, MMP: matrix metalloproteinase, TGFβ: transforming growth factor beta, PDGFRb: pericytes express platelet-derived growth factor receptor beta, CSF: cerebrospinal fluid, sPDGFR: soluble PDGFR, CAMs: cell adhesion molecules, s4-1BBL: soluble 4-1BB ligand, PECAM-1: platelet endothelial cell adhesion molecule 1, VEGFA: vascular endothelial growth factor A, SPARC: secreted protein acidic and rich in cysteine, N/A: not applicable.

Factors	Disease	Expressions or Levels	Phenotype	Reference
Glucose transporter (GLUT1)	AD	Decrease	Decreased in glucose concentrations in the CNS	[139]
LRP1	AD	Decrease	Decreased with increasing oxidative stress	[130]
	AD	Decrease	Decreased with increasing oxidative stress	[131]
LRP1 and Pgp	AD mice	Decrease	Hinders amyloid clearance from the brain	[132]
RAGE	AD	Increase	Elevated in brain endothelial cells, promoting neuronal inflammation	[136]
Aquaporin-4 (AQP4)	AD	Decrease	Increasing neurofibrillary tangles	[109]
	AD	Decrease	Increased amyloid-b eta pathology	[109]
Apolipoprotein E 2/3 (ApoE 2/3)	Maintain BBB integrity	Increase	Interacts with LRP-1 in pericytes to block the “CypA-NF-kb-MMP-9” pathway, leading to the inhibition of MMPs	[111]
APOE4	AD	Increase	Shifting the rapid clearance of soluble Ab40/42 by LRP1 to the slower clearance by VLDLR	[167]
	AD	Increase	Reduced BBB integrity by promoting pericyte degeneration	[111]
	Endothelial LRP1 knockout mice	N/A	CypA-NFkB-MMP-9 activation	[173]
	Endothelial LRP1 knockout mice	N/A	TJ and BBB damage	[173]
	APOE4-transfected mice	Increase	Inhibits the expression of Glut1	[37]
	APOE4-transfected mice	Increase	Increase expression of RAGE	[37]
	AD patients	Increase	More prone to BBB damage	[168]
	AD patients	Increase	Reduced pericytes	[130]
TGFβ	AD transgenic mice	Increase	Abundantly expressed in astrocytes	[175]
	AD transgenic mice	Increase	Promote amyloidosis	[175]
	AD transgenic mice	Increase	Promote amyloid clearance by microglia	[176]
Platelet-derived growth factor receptor beta (PDGFRb)	Aging	Increase	Elevated in cerebrospinal fluid (CSF)	[178]
	BBB damage	Increase	Impairment of pericytes	[178]
Soluble PDGFR (sPDGFR)		Increase	Pericyte and blood–brain barrier disruption	[181]
	Pdgfr +/− pericyte-deficient mice	Increase	BBB impairment caused by neuronal degeneration	[48]
	AD patients	Increase	Leakage of the BBB in the hippocampus	[15]
	AD patients	Increase	Increased soluble PDGFRb (sPDGFRb) in cerebrospinal fluid	[182]
	Neurodegenerative diseases	Increase	As a biomarker in cerebrospinal fluid	[179]
s100b	Stressed BBB	Increase	Protein biomarker produced by astrocyte endfeet	[190]
CAMs, zonulin, and s4-1BBL	BBB impairment	N/A	As biomarkers	[189]
PECAM-1, P-selectin, and E-selectin	BBB impairment	Increase	As biomarkers	[189]
RepSox	Neurological diseases	N/A	Inhibits TGF-B, VEGFA, and inflammation-related pathways	[191]
	Neurological diseases	N/A	Increases BBB resistance and induces TJs and transporters	[191]
	Neurological diseases	N/A	Reduces paracellular permeability by activating Notch and Wnt pathways	[191]
SPARC	AD	N/A	Reduces transendothelial electrical resistance (TEER) and TJs	[188]
	AD	N/A	The SPARC–collagen binding domain can be a therapeutic target in AD	[188]
	AD	N/A	SPARC/Hevin can be a therapeutic target for modulating AD progression	[187]

## 11. Discussion

Thirteen AQP4 paralogs have currently been identified in higher mammals, and their functions were shown as transporting water across biological membranes or the diffusion of small solutes. Among these, AQP2, 4, and 5 were classical water channels, while AQP0, 1, and 6 have dual roles, transporting water and ions. The major functions of AQPs are (1) fluid homeostasis for urine or water and secretion in the lungs, (2) signal transduction for skeletal muscle contraction for dorsal root ganglion axonal growth and sensor function for balance in the inner ear or the integrity of eye lens, (3) the integrity of the BBB, and (4) cell motility for migration at the differential stages of polarization, adhesion, and retraction [193]. The glymphatic system, associated with AQP4, was discovered in 2012, and its function was removing the waste products from the brain, preventing these wastes from causing neurodegeneration [145]. Conditional KO or the overexpression of AQP4 decreases or increases water uptake, respectively [160,194]. It has been well-documented that AQP4 plays an important role in brain edema formation under ischemia and trauma injury conditions. Interestingly, the endfoot membrane of AQP4 was increased, but the total protein levels of it had no changes, implying that the relocalization of this molecule occurred under stroke and traumatic brain injury [152]. Moreover, hypoxia also induced the relocalization of AQP4 from intracellular vesicle pools to the cell surface membrane [55]. 

The administration of a calmodulin inhibitor, trifluoperzaine, inhibits its relocalization to protect against brain edema. Furthermore, AQP4 is associated with BBB integrity via interactions with cytoskeleton proteins, such as agrin [195], dystrophin [196], or alpha-syntrophin [197], to maintain the polarization of astrocytes. Astrocytes lacking these proteins showed the downregulation of AQP4 function. For example, the downregulation of the dystrophin-associated protein complex (DAPC) affects AQP4 anchoring and BBB integrity. Furthermore, Kir4.1, the inward rectifying potassium channel, was colocalized with AQP4 in the astrocytic endfeet. The interaction was connected by alpha-syntrophin [198]. A decrease in dystophin and mislocalization of AQP4 were found in patients with epilepsy [199]. Thus, the deregulation of the astrocytic polarization caused by these cytoskeletal proteins leads to a loss in AQP4 function in maintaining the integrity of the BBB [200].

The mammalian AQPs exert their effect on providing dynamic fluid homeostasis via different signaling pathways, such as (1) channel gating, including protonation or CaM binding, or (2) subcellular relocalization. A low pH causes the protonation of histidine, leading to conformational changes in channels followed by pore occlusion intracellularly [201]. This Ca^2+^ sensitivity interacts with CaM to regulate water permeability via AQP0 [202,203]. AQPs are trafficked from intracellular vesicles to the plasma membrane via Ca^2+^-calmodulin- or phosphorylation-dependent mechanisms [200]. Changes in the expression of AQPs, leading to changes in their protein levels, could affect the permeability of the BBB; however, this takes a lot of time compared to the trafficking of these receptors from intracellular vesicles to the plasma membrane.

Some methods have recently been developed and provided for studying cell migration and BBB dynamics. The human BBB organoids model could tightly maintain endothelial cells’ lining; however, this model, lacking parameters in flow and microvessel-like structures, which cannot be architectured in organoids, rather generates spheres [204]. On the other hand, the in vitro models of two-dimensional (2D) transwell systems have generally been used for migration, barrier permeability, or transepithelial/transendothelial electrical resistance (TEER) [205,206], but they also lack blood flow conditions similar to those of BBB organoids model. Organs-on-a-chip of BBB, 3D microfluidic, and brain microvessel models have been developed to overcome these limitations [207,208,209,210]. They mimic the brain BBB niches to provide shear stress, unidirectional flow, cell–cell interactions, receptor-mediated transcytosis, extracellular matrix environment, etc. Alternatively, they could collaborate with live 3D tacking or advanced imaging using spinning disk confocal or lattice light sheet microscopy (LLSM) to assist in studying the molecular mechanisms in the BBB [211].

Alzheimer’s disease, Parkinson’s disease, and Huntington’s disease are devastating, incurable, and debilitating neurodegenerative diseases. There are unmet and urgent needs for developing therapeutic drugs because there are no effective disease-modifying therapies that target their underlying pathological mechanisms. Using computer-aided drug design (CADD), pharmacologists could reduce the time-consuming and expensive processes by about 10–15 years via the determination of protein structure through CADD and reducing the traditional series steps, which include nuclear magnetic resonance (NMR) spectroscopy, X-ray crystallography, cryo-EM, homology modeling, the identification of binding sites, docking studies, virtual screening, quantitative structure, activity relationship studies, and pharmacophore modeling [212]. Through recent advances in CADD, combined with software for them, candidate drugs could be selected in an easier manner than before, and they could be further clarified with optimal platforms through efficient high-throughput screening (HTS). The steps of HTS include drug identification for higher predictability and clinical applicability via the steps of target recognition, the management of compounds, preparation of reagent, development of assay as well as screening itself [213]. Before lead identification and drug lead optimization with combinatorial and medicinal chemistry, hits from HTS will show the prototypes. HTS reduces the risk of candidates failing in preclinical- and clinical-related research. Therefore, recent advances in HTS, combined with computational tools, are expected to cost less and discover the CNS drugs targeting these neurodegenerative diseases faster in the future.

## 12. Conclusions

The BBB comprises cells that tightly regulate the normal microenvironment and neuronal activity. Any physiological or molecular disruption of these cells results in the breakdown of the BBB. Aging accelerates BBB breakdown. Various physiological properties of the BBB are impaired during aging, leading to BBB dysfunction. The infiltration of neurotoxins into the brain may cause cognitive impairment and neurodegeneration. BBB damage may further lead to dementia, including AD and VaD. Following BBB disruption in AD, the deposition of amyloids and NFTs in blood vessels leads to further inflammation of the NVU and neurodegeneration associated with cognitive decline. Therefore, BBB leakage may serve as a new biomarker of various neurological disorders, such as AD, VaD, and other diseases associated with cognitive decline.

The disruption of the BBB in the pathogenesis of cognitive impairment has been associated with normal aging and dementia; nonetheless, further studies are warranted to identify the cellular and molecular mechanisms underlying BBB maintenance and BBB breakdown as well as repair in neurodegenerative and neurocognitive diseases.

## Figures and Tables

**Figure 1 ijms-24-02909-f001:**
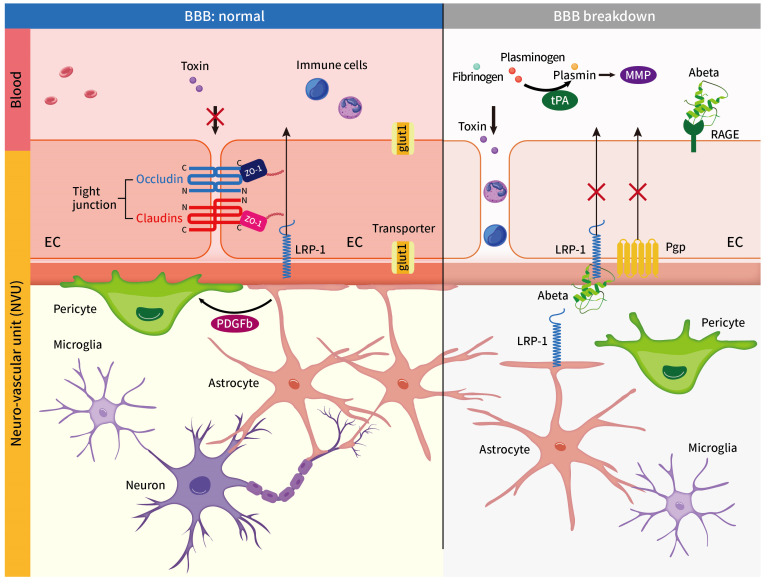
The interaction of differential cells, molecular factors, and transporters in the BBB under physiological (normal) or pathological (BBB breakdown) conditions. (BBB: blood–brain barrier, GLUT1: glucose transporter, LRP1: low-density-lipoprotein (LDL)-receptor-related protein 1, Pgp: phosphoglycolate phosphatase, RAGE: advanced glycosylation end product (AGE) receptor, MMP: matrix metalloproteinase, and PDGFRb: pericytes express platelet-derived growth factor receptor beta).

## Data Availability

No new data were created or analyzed in this study. Data sharing is not applicable to this article.

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
