# Peer review of "Endothelial Dysfunction in Neurodegenerative Diseases"

_ijms, 2023, doi:10.3390/ijms24032909_

Round 1
Reviewer 1 Report
This article has the merit of summarising the anatomical, functional and biochemical structures of the BBB, taking into account the most recent research.
Author Response
Dear reviewer:
Thank you very much for your appreciation and valuable advice.
Reviewer 2 Report
Dear Editor,
The manuscript by Fang et al. discusses the current understanding and recent advances vascular dysfunction in aging and neurodegenerative diseases, particularly AD and vascular dementia.
The work is comprehensive, informative, nicely-written, timely and up-to-date (in most parts). Authors were successful in providing some well compiled opinions, summaries and nice figures which make this work a good starting point for researchers interested in in aging and neurodegenerative diseases.
However, there is a number of major and minor points that would need to be addressed in order to improve the quality of this paper before it can be accepted for publication.
General:
- This work overlooked some essential and up-to-date work regarding the recent advances in glial and BBB target validation and future therapies. I have made some suggestions below but author is encouraged to consider citing updated references throughout the work, whenever possible.
Major:
-Astrocytes line 227: the authors omit a key study from Kitchen et al 2020 demonstrating that the development of edema following injury-induced hypoxia is AQP4 dependent. That study shows that CNS edema is associated with increases both in total aquaporin-4 expression and aquaporin-4 subcellular translocation to the BBB and blood-spinal-cord-barrier (BSCB). Pharmacological inhibition of AQP-4 translocation to the BSCB prevents the development of CNS edema and promotes functional recovery in injured rats.
This role has been recently been confirmed by the work of Sylvain et al BBA 2021 which has demonstrated that targeting AQP4 effectively reduces cerebral edema during the early acute phase in in stroke using photothrombotic stroke model. They have also shown a link to brain energy metabolism as indicated by the increase of glycogen levels. References to be included:
https://www.cell.com/cell/fulltext/S0092-8674(20)30330-5.
https://pubmed.ncbi.nlm.nih.gov/33561476/
-Line 360: Hypoxia has known to induce apoptotic pathway in human astrocytes. Reference:
https://pubmed.ncbi.nlm.nih.gov/29311824/
Moreover, Figure 1 needs to be updated in the light of the recent work by Oscar Ndunge. Reference:
https://www.ncbi.nlm.nih.gov/pmc/articles/PMC9798958/
-Channel proteins line 403: The glymphatic function has been implicated in brain waste clearance and neurodegenerative diseases. Though the field tends to discuss ‘water’ and ‘fluid’ in a manner that incorrectly suggests their interchangeability. However, it is important to note that in the glymphatic system, the clearance of brain waste occurs through paracellular flow. Classic tracer studies measure this paracellular flow, while the use of H217O captures both paracellular flow and diffusive transcellular exchange of water. Importantly, both are AQP4 dependent — one directly and one indirectly. This needs to be clarified. Reference:
https://www.nature.com/articles/s41583-021-00514-z
https://academic.oup.com/brain/article/145/1/64/6367770
-Line 413-414 “Furthermore, disruption of AQP4 expression results in Aβ deposition and inflammation in the human brain, leading to AD”: The increased AQP expression and the redistribution/surface localization can be two different concepts. For example, previous studies have shown an increased in AQP4 membrane localisation in primary human astrocytes which wasn’t accompanied by a change in AQP4 protein expression levels. This mislocalization can be a potential therapeutic target. References:
https://www.ncbi.nlm.nih.gov/pmc/articles/PMC5765450/
https://www.ncbi.nlm.nih.gov/pubmed/31242419
https://pubmed.ncbi.nlm.nih.gov/26013827/
-Line 416-417 “Therefore, AQP4 may be a promising therapeutic target in AD and other diseases of the CNS”: AQPs have been validated as an important drug target but there is no single drug that has yet been approved to successfully target it. This needs to be mentioned as it highlights the importance for studies similar to the current one by providing alternative routes of targeting AQP function compared to the traditional pore-blocking approach. References to be included:
https://pubmed.ncbi.nlm.nih.gov/34863533/
https://www.ncbi.nlm.nih.gov/pmc/articles/PMC6480248/
Based on the above and towards the end of discussion, authors need to discuss recent trends in targeting the molecular and signalling mechanisms of AQPs rather than just the traditional approaches. The importance of this new approach has been discussed in these references which should be included to enrich the discussion of current manuscript. References:
https://pubmed.ncbi.nlm.nih.gov/34973181/
https://www.mdpi.com/1422-0067/23/3/1388
Minor:
- Authors need to briefly discuss future directions following towards the end of their discussion and conclusion. This could include, but not limit to, the use of humanized self-organized models, organoids, 3D cultures and human microvessel-on-a-chip platforms especially those which are amenable for advanced imaging such as TEM and expansion microscopy since they enable real-time monitoring of cell invasion and BBB dynamics. References to be included:
https://pubmed.ncbi.nlm.nih.gov/33117784/
https://pubmed.ncbi.nlm.nih.gov/30165870/
-Neurodegenerative diseases are yet incurable diseases. Author needs to point out to the recent advances in applying the use of high-throughput screening and computer-aided drug design as have been nicely reviewed by Aldewachi et al 2021 as they can provide a novel insight that can support AQP target validation in future studies. References to be included:
https://pubmed.ncbi.nlm.nih.gov/33925236/
https://pubmed.ncbi.nlm.nih.gov/33672148/
Best.
Author Response
Major:
-Astrocytes line 227: the authors omit a key study from Kitchen et al 2020 demonstrating that the development of edema following injury-induced hypoxia is AQP4 dependent. That study shows that CNS edema is associated with increases both in total aquaporin-4 expression and aquaporin-4 subcellular translocation to the BBB and blood-spinal-cord-barrier (BSCB). Pharmacological inhibition of AQP-4 translocation to the BSCB prevents the development of CNS edema and promotes functional recovery in injured rats.
This role has been recently been confirmed by the work of Sylvain et al BBA 2021 which has demonstrated that targeting AQP4 effectively reduces cerebral edema during the early acute phase in in stroke using photothrombotic stroke model. They have also shown a link to brain energy metabolism as indicated by the increase of glycogen levels. References to be included:
https://www.cell.com/cell/fulltext/S0092-8674(20)30330-5.
https://pubmed.ncbi.nlm.nih.gov/33561476/
Response:
Thank you very much for the valuable advice.
We have added this important information into our revised manuscript. Studies have shown that hypoxia leads to subcellular relocalization of AQP4 to the surface of BBB and blood-spinal-cord-barrier (BSCB), accompanied by increasing water permeability via a signaling pathway associated with calmodulin (CaM). In vivo evidence showed that inhibition of AQP4 subcellular localization to the BSCB reduces spinal cord water content, the extent of subsequent injury and enhances the recovery in a rat spinal cord injury model of CNS edema. (Kitchen et al., 2020). Moreover, targeting CaM to inhibit the cerebral edema mediated by AQP4 may provide the therapeutic target during the acute phase in the photothrombotic stroke model (Sylvain et al., 2021). (page 9, line 238)
-Line 360: Hypoxia has known to induce apoptotic pathway in human astrocytes. Reference:
https://pubmed.ncbi.nlm.nih.gov/29311824/
Response: Thank you very much for the valuable advice. We have added this important information into our revised manuscript. By use of gene expression microarrays and proteome profiling array, hypoxia was found to induce the Wnt and p53 related apoptotic signaling pathways
in primary human astrocytes (Salman et al., 2017). (page 17, line 366)
Moreover, Figure 1 needs to be updated in the light of the recent work by Oscar Ndunge. Reference:
https://www.ncbi.nlm.nih.gov/pmc/articles/PMC9798958/
Response: Thank you very much for the valuable advice.We have added this important information into our revised manuscript. More recently, vascular endothelial cell function is also disrupted by cerebral malaria (CM) infection. (Akide Ndunge et al., 2022). A hallmark of endothlial dysfunction is adhesion of infected RBC, along with platelets and immune cells to the vascular endothelium to sequester in the patient's brain upon infection with the plasmodium parasite. Disruption of BBB integrity and recruitment of peripheral immune cells result from iRBC cell adhesion to the brain microvasculature, especially endothelial cells, during the trophoblast and schizont stages which induce vasospasm. It also changes in the levels of vascular regulatory molecules further promotes pro-inflammatory and prothrombotic states. The sequestration of iRBCs in the brain occurs via cell adhesion molecules, ICAM-1 and endothelial protein C receptor (EPCR), expressed on the surface of brain ECs. It induces monocyte to phagocyte iRBCs and then excretes the parasite and plasmodium toxins. All of these consequences cause endothelial cell dysfunction which contributes to activation of immune cells, axonal damage, neurodegeneration and cognitive impairment. (page 23 line 587)
-Channel proteins line 403: The glymphatic function has been implicated in brain waste clearance and neurodegenerative diseases. Though the field tends to discuss ‘water’ and ‘fluid’ in a manner that incorrectly suggests their interchangeability. However, it is important to note that in the glymphatic system, the clearance of brain waste occurs through paracellular flow. Classic tracer studies measure this paracellular flow, while the use of H217O captures both paracellular flow and diffusive transcellular exchange of water. Importantly, both are AQP4 dependent — one directly and one indirectly. This needs to be clarified. Reference:
https://www.nature.com/articles/s41583-021-00514-z
https://academic.oup.com/brain/article/145/1/64/6367770
Response: Thank you very much for the valuable advice. We have added this important information into our revised manuscript. AQP4 is found in endfeet of astrocytes, ventricle lining, tripartite synapses, and the surrounding meninges. A tight correlation between AQP4 and glymphatic function has been obtained from the studies of Aqp4 or alpha-syntrophin (Snta1) knockout (KO) mice. The previous studies found that dynamic translocation of AQP4, from intracellular vesicle pools to the plasma membrane is responsible to water homeostasis in health and disease. (Salman et al., 2022, Ciappelloni, S. et al2019 Lisjak et al., 2017 Salman et al., 2021). In pathological conditions, pharmacological inhibition of AQP4 relocates to astrocyte endfoot to promote edema formation, and Salman et al proposed that AQP4 is directly involved in this process (Salman et al., 2021).
However, there is still no conclusion on whether the brain water homeostasis is achieved by changing the AQP4 itself directly or whether it is an indirect effect caused by genetic deletion of AQP4 (MacAula, 2021). Aqp4 KO mice exhibit a series of physiological and cellular changes, including increase in brain water content (Haj-Yasein, N. N.), extracellular space in brain tissue (Yao, 2008), expression of excitatory amino acid transport protein 2 (EAAT2) (Zeng 2007) and decrease in expression of the anchor proteins, α-syntrophin and dystrophin in astrocyte terminals (Eilert-Olsen 2012), which all may provide clues to affect the expression of other membrane proteins. (page 19 line 444)
-Line 413-414 “Furthermore, disruption of AQP4 expression results in Aβ deposition and inflammation in the human brain, leading to AD”: The increased AQP expression and the redistribution/surface localization can be two different concepts. For example, previous studies have shown an increased in AQP4 membrane localisation in primary human astrocytes which wasn’t accompanied by a change in AQP4 protein expression levels. This mislocalization can be a potential therapeutic target. References:
https://www.ncbi.nlm.nih.gov/pmc/articles/PMC5765450/
https://www.ncbi.nlm.nih.gov/pubmed/31242419
https://pubmed.ncbi.nlm.nih.gov/26013827/
Response: Thank you very much for the valuable advice. We have added this important information into our revised manuscript. However, the protein levels of AQP4 instead of subcellular translocation were not changed in many scenario, including hypothermia via transient receptor potential vanilloid 4 (TRPV4) activation (Salman et al., 2017), osmolality changes via PKA phosphorylation in primary human cortical astrocyte (Kitchen et al., 2015). Moreover, disturbance of dynamic redistribution in AQP4 could affect the process and synaptic activity in astroctye. (Ciappelloni et al., 2019). (page 18 line 424)
-Line 416-417 “Therefore, AQP4 may be a promising therapeutic target in AD and other diseases of the CNS”: AQPs have been validated as an important drug target but there is no single drug that has yet been approved to successfully target it. This needs to be mentioned as it highlights the importance for studies similar to the current one by providing alternative routes of targeting AQP function compared to the traditional pore-blocking approach. References to be included:
https://pubmed.ncbi.nlm.nih.gov/34863533/
https://www.ncbi.nlm.nih.gov/pmc/articles/PMC6480248/
Response: Thank you very much for the valuable advice. We have added this important information into our revised manuscript. There is no effective drug targeting AQP4 directly approved for clinical use, although AQP4 has been proven to have the therapeutic potential in stroke, traumatic brain injury and other neurodegenerative diseases (Salman et al., 2022) ( Abir-Awan et al., 2019). Targeting the calmodulin-mediated AQP4 phosphorylation and translocation in astrocytes with antipsychoic drug, trifluoperazine, ablated CNS edema in spinal cord injury model, and this provides the potential for specificity to reduce the side effect (Kitchen et al. 2020). There are also differential kinases, for example, ERK, p38, calmodulin-dependent protein kinase, implicated in AQP4 relocalization, but the AQP4 is phosphorylated by these kinases directly remaining to be determined (Salman et al., 2017a; salman et al., 2017b). Although their pores are intrinsically undruggable or could be blocked by other small molecules remain to be studied; however, targeting the molecular mechanism of them provide new strategies for drug discovery. The selective antagonists or blockers of AQP4 water pore has not been proven, however, the AQP4 inhibitor, TGN-020 or AER-270 have the beneficial effect in ischemia-induced brain edema (Igarashi et al., 2011, Farr et al., 2019). The protective effect of these druges or molecules may have other AQP-4 function independently. (page 19 line 430)
Based on the above and towards the end of discussion, authors need to discuss recent trends in targeting the molecular and signalling mechanisms of AQPs rather than just the traditional approaches. The importance of this new approach has been discussed in these references which should be included to enrich the discussion of current manuscript. References:
https://pubmed.ncbi.nlm.nih.gov/34973181/
https://www.mdpi.com/1422-0067/23/3/1388
Response: Thank you very much for the valuable advice. We have added this important information into our revised manuscript.
13 AQP4 paralogs have currently been identified in higher mammals, and their functions were shown in transporting water across biological membranes or diffusion of small solutes. Among these, AQP2, 4, and 5 were classical water channels, and AQP0, 1, and 6 have dual roles in transport of water and ions. The major functions of AQPs are (1) fluid homeostasis for urine or water and secretion in lungs, (2) signal transduction for skeletal muscle contraction for dorsal root ganglion axonal growth and sensor function for balance in the inner ear or integrity of eye lens, (3) integrity of BBB, (4) cell motility for migration at the differential stages of polarization, adhesion, and retraction (Wagner et al., 2022). The glymphatic system, associated with AQP4, was discovered in 2012, and its function was removing the waste products from the brain, preventing these wastes to cause the neurodegeneration (Iliff et al., 2012). Conditional KO or overexpression of AQP4 decreases or increases water uptake respectively (Haj-Yasein et al., 2011, Yang et al., 2008). It has been well documented that AQP4 plays an important role in brain edema formation under ischemia and trauma injury condition. Interestingly, the endfoot membrane of AQP4 was increased, but the total protein levels of them have no changes, implying that the re-localization of this molecule occurred under stroke and traumatic brain injury (Kitchen et al., 2018). Moreover, hypoxia also induced re-localization of AQP4 from intracellular vesicle pools to the cell surface membrane (Kitchen et al., 2020).
The administration of calmodulin inhibitor, trifluoperzaine, inhibits its relocalization to protect against brain edema. Furthermore, AQP4 is associated with the BBB integrity by interaction with cytoskeleton proteins, for example, agrin (Berzin et al., 2000), dystrophin (Vajda et al., 2002) or alpha-syntrophin (Dmytrenko et al., 2003) to maintain the polarization of astrocytes. Astrocytes lacking these proteins showed the down-regulation of AQP4 function. For example, down-regulation of dystrophin-associated protein complex (DAPC) affects the AQP4 anchoring and BBB integrity. Furthermore, Kir4.1, the inward rectifying potassium channel, was co-localized with AQP4 in the astrocytic endfeet. The interaction was connected by alpha-syntrophin (Connors et al., 2004). Increase in K+ concentration could elevate the protein levels of AQP4 and decrease the dystophin in patients with epilepsy (Alvestad et al., 2013). Thus, the deregulation of astrocytic polarization caused by these cytoskeletal proteins leads to loss of AQP4 function in maintaining the integrity of BBB (Markou et al., 2022).
The mammalian AQPs exert their effect in providing dynamic fluid homeostasis via different signaling pathways, for example, (1) channel gating including protonation or CaM binding or (2) subcellular relocalization. The low pH causes protonation of histidine, leading to conformational changes in channels followed by pore occlusion intracellularly (Gotfryd et al., 2018). This Ca2+ sensitivity interact with CaM to regulate water permeability via AQP0 (Nemeth-Cahalan, 2013, Reichow et al. 2013). AQPs are trafficking from intracellular vesicles to the plasma membrane via Ca2+-calmodulin- or phosphorylation-dependent mechanisms (Markou et al., 2022). Changes of expression of AQPs, leading to changes in their protein levels could affect the permeability of BBB; however, it takes much time compared to the trafficking these receptors from intracellular vesicle to the plasma membrane. (page 27 line 621)
Minor:
- Authors need to briefly discuss future directions following towards the end of their discussion and conclusion. This could include, but not limit to, the use of humanized self-organized models, organoids, 3D cultures and human microvessel-on-a-chip platforms especially those which are amenable for advanced imaging such as TEM and expansion microscopy since they enable real-time monitoring of cell invasion and BBB dynamics. References to be included:
https://pubmed.ncbi.nlm.nih.gov/33117784/
https://pubmed.ncbi.nlm.nih.gov/30165870/
Response:
Thank you very much for the valuable advice. We have added this important information into our revised manuscript.
Recently, some methods were developed and provided for study the cell migration and BBB dynamics. The human BBB organoids model could maintain endothelial cells lining tightly. However, this model, lacking parameters in flow and microvessel-like structures which cannot be architectured in organoids, rather spheres are generated (Urich et al., 2013). On the other hand, the in vitro models of two-dimensional (2D) transwell system have generally been used for migration, barrier permeability, or transepithelial/transendothelial electrical resistance (TEER) (Biegel and Pachter, 1994; He et al., 2014), but it also lacks blood flow conditions like organoids. The organ-on-a-chip of BBB, 3D microfluidic and brain microvessel models have been developed to overcome these limitations (van Der Helm et al., 2016; Wevers et al., 2018; Oddo et al., 2019; Park et al., 2019). It mimics the brain BBB niches to provide shear stress, unidirectional flow, cell-cell interactions, receptor-mediated transcytosis, and extracellular matrix environment, etc. Otherwise, it could collaborate with live 3D tacking or advanced imaging using spinning disk confocal or lattice light sheet microscopy (LLSM) assisting to study the molecular mechanisms in BBB (Salman et al., 2020). (page 28 line 660)
-Neurodegenerative diseases are yet incurable diseases. Author needs to point out to the recent advances in applying the use of high-throughput screening and computer-aided drug design as have been nicely reviewed by Aldewachi et al 2021 as they can provide a novel insight that can support AQP target validation in future studies. References to be included:
https://pubmed.ncbi.nlm.nih.gov/33925236/
https://pubmed.ncbi.nlm.nih.gov/33672148/
Response: Thank you very much for the valuable advice. We put this paragraph in the discussion and conclusion. Alzheimer’s disease, Parkinson’s disease, or Huntington’s disease, are devastating, incurable, and debilitating neurodegenerative diseases. There are unmet and urgent needs to develop therapeutic drugs because there is no effctive disease-modifying therapy targeting their underlying pathological mechanisms. Using computer-aided drug design (CADD), pharmacologists could save the time-consuming and expensive processes by about 10-15 years via determination of the protein structure by CADD and reducing the traditional series steps which include nuclear magnetic resonance (NMR) spectroscopy, x-ray crystallography, or cryo-EM, homology modelling, identification of binding sites, docking studies, virtual screening, quantitative structure, activity relationship study, and pharmacophore modelling (Salman et al., 2021). By recent advances in CADD, combined with software for them, the candidate drugs could be selected easier than before and they could be further clarified with the optimal platforms by efficient high-throughput screening (HTS). The steps of HTS include drug indentification for higher predictability and clinical applicability via the steps of target recognition, management of compounds, as well as the screening itself (Aldewachi et al., 2021). Before the lead identification and drug lead optimization with the combinatorial and medicinal chemistry, hits from the HTS screening will show the prototypes. The HTS reduces the risk of candidates failure for pre-clinical and clinical related research. Therefore, recent advances in HTS combined with computational tools are expected to less cost and faster to discover the CNS drugs targeting these neurodegenerative diseases in the future. (page 29 line 673)
Round 2
Reviewer 2 Report
Dear Editor,
The authors have successfully addressed the majority of my comments and concerns in order to improve the quality of the manuscript.
I believe that the new sections, improved ones, and updated references, have contributed to enhancing the clarity of the manuscript, which I can now endorse for publication.
All the best!